

**Litter decomposition enhances volatile organic compound emission from a freshwater**
**wetland: insights from year-round in situ field experiments**
Hua Fang[1,2], Ting Wu[1,2]* Shutan Ma[1,2], Qina, Jia[1], Fengyu Zan[1,2], Juan Zhao[1,2], Jintao Zhang[1],
Zhi Yang[1], Hongling Xu[1], Yuzhe, Huang[1], Xinming Wang[3,4]*
[1] School of Ecology and Environment, Anhui Normal University, Wuhu, 241000, China.
[2] Center of Cooperative Innovation for Recovery and Reconstruction of Degraded Ecosystem
in Wanjiang City Belt, Wuhu, 241000, China.
[3] State Key Laboratory of Organic Geochemistry and Guangdong Key Laboratory of
Environmental Protection and Resources Utilization, Guangzhou Institute of Geochemistry,
Chinese Academy of Sciences, Guangzhou 510640, China
[4] CAS Center for Excellence in Deep Earth Science, Guangzhou 510640, China
*Corresponding author:
Dr. Ting Wu (wuting19@mail.ahnu.edu.cn)
Dr. Xinming Wang (wangxm@gig.ac.cn)





**Abstract**
Plant litter could be a potential source of atmospheric volatile organic compounds (VOCs).
Previous studies are mostly restricted to forest litter, but VOC budget of wetlands, especially
freshwater wetlands, resulting from litter decomposition remains largely unexplored. Here we
performed in-situ VOC flux measurements in a freshwater wetland and three treatments
including A (no litter addition), B (1.4 kg litter) and C (2.8 kg litter) were designed to
investigate impacts of litter decomposition on wetland-atmosphere exchange of VOCs.
During year-round litter decomposition, average fluxes of net VOCs for B and C were
$5.93\pm3.13$ μg m$^{-2}$ h$^{-1}$ and $8.30\pm4.00$ μg m$^{-2}$ h$^{-1}$, respectively, significantly higher than that of A
($2.90\pm2.74$ μg m$^{-2}$ h$^{-1}$). These results suggested that freshwater wetland was a potential source
of atmospheric VOCs and litter decomposition enhanced VOC release. Net VOC flux showed
clear seasonal patterns and was highly correlated with ambient temperature ($p<0.05$). In
general, higher positive VOC fluxes were observed in hot summer, while lower positive VOC
fluxes or negative VOC fluxes were observed in cold winter. Moreover, the release (positive
flux) or uptake (negative flux) of VOCs varied to chemical groups. Specifically, non-methane
hydrocarbons (NMHCs) including alkanes, alkenes and aromatics showed positive net fluxes,
and increased with added litter. Halocarbons showed a negative net flux in A, but positive net
fluxes in B and C. While oxygenated volatile organic compounds (OVOCs) showed negative
net fluxes in both A and B, and switched to a positive flux in C. Positive net fluxes of volatile
organic sulfide compounds (VOSCs) were observed in three treatments. According to flux
variations of specific VOC group, it has been suggested that temperature-driven biotic and
abiotic processes co-modulated VOC release or uptake occurring in the freshwater wetland.

**Keywords:** Biogenic volatile organic compounds; Fluxes; Litter decomposition; Freshwater
wetland; Biosphere-atmosphere exchange







**1. Introduction**
Volatile organic compounds (VOCs) play an important role in atmospheric chemistry of
troposphere (Toro et al., 2006; Schnell et al., 2009; Chang et al., 2022). Once VOCs are
emitted into real atmosphere, they can react with various oxidants (e.g., OH, $HO_2$ and $RO_2$) to
form ozone ($O_3$) and secondary organic aerosols (SOA) (Kalashnikov et al., 2022),
influencing regional air quality (Li et al., 2019) and climate change (Scott et al., 2014).
Although anthropogenic contribution dominates the VOC mass in urban environments
(Lamarque et al., 2010), ~ 90% of VOCs is actually from the biogenic source, namely
biogenic VOCs (BVOCs), on a global scale (Otter et al., 2003). In addition to the
contributions to $O_3$ and SOA, BVOCs have been revealed to significantly affect natural
ecosystems (Fowler et al., 2009; Park et al., 2013). About 1,000 Tg/year of carbon was
released into the atmosphere from ecosystems in the form of VOCs (Guenther et al., 2012).
These BVOCs could change the specific biogeochemical processes within natural
environments through stimulating or inhibiting the growth of plants and microbes (Peñuelas
et al., 2014), modulating the cycling of carbon and nitrogen (Smolander et al., 2006; Gray et
al., 2014).
Considering their great impacts on atmosphere and natural ecosystems, numerous studies on
BVOCs have been conducted for decades (Duan et al., 2020). Most of these studies
investigated living vegetation and have reported them as important contributors to
atmospheric BVOCs. For example, terpenes, mainly isoprene, monoterpenes and
sesquiterpenes, emitted by living terrestrial vegetations could account for over 40% of BVOC
emissions (Navarro et al., 2014). Compared to living vegetation, studies on VOCs emitted by
dead litters, particularly during their long-term decomposition, have been quite limited.
Notably, in recent years a growing number of studies have found that the decomposition of
dead litter could also release significant VOCs, changing seasonal profile of local or even
regional VOCs (Leff and Fierer, 2008; White et al., 2009; Greenberg et al., 2012).
Furthermore, Gray et al. (2010) reported that the percentage of OVOCs such as methanol
could reach 99% of emitted VOCs during litter decomposition (litter types: deciduous species,
evergreen species and dead grass leaves). While Faiola et al. (2014) found that monoterpenes
(80.3%) dominated the total VOCs released from leaf litters of coniferous forest. These



different research results reflected that the composition of VOCs emitted by litters varied
significantly to litter types. In addition, temperature, sunlight, moisture and microbial
metabolism have been identified as the key factors affecting uptake or release of VOCs during
litter decomposition (Niinemets et al., 2004; Vickers et al., 2009; Zhang et al., 2021; Zeng et
al., 2022). However, so far these studies focused on forest litter (Leff et al., 2008; Isidorov et
al., 2010; Greenberg et al., 2012; Gray et al., 2014; Faiola et al., 2014; Svendsen et al., 2018)
and terrestrial environments (Breider et al., 2015). VOCs emitted during the decomposition of
litters from other natural environments are still largely unexplored, inducing the uncertainty
of estimating BVOC budget by models (Tang et al., 2019).
Wetlands cover only 5%-8% of the land surface but have a disproportionate impact on the
global carbon cycle (William et al., 2013; Davidson et al., 2018; Villa et al., 2019;Anderson
et al., 2020). Numerous studies concern wetlands mainly due to their well-known roles as the
sink of $CO_2$ and the source of $CH_4$ (Peng et al., 2022). As previously reported, the global net
carbon sequestration from wetlands was 830 Tg/year and meanwhile they also contributed
over 20% of global $CH_4$ emission (Whalen et al., 2005; Bergamaschi et al., 2007; Bloom et al.,
2010). However, far less wok has been devoted to investigating uptake or release of VOCs in
these areas. Although some studies have recently reported VOC fluxes in peat, fen and forest
wetlands (Haapanala et al., 2006; Hellén et al., 2018; Jiao et al., 2018; Seco et al., 2020;
Männistö et al., 2023), the role of freshwater wetlands in VOC budget is still poorly
understood. Moreover, the decomposition of plant litters occurring in the freshwater wetland
is a very common natural phenomenon, but how this process affects the uptake or release of
VOCs remains unanswered.
Herein, we conducted in situ field experiments to measure VOC fluxes during year-round
litter decomposition in a freshwater wetland. The entire observation consisted of 11
campaigns from January 9 to December 14, 2022. A total of 103 valid samples were obtained
during the observation. To the best of knowledge, this is the first-ever result on VOC flux
measurements in a freshwater wetland under long-term period of litter decomposition (across
four seasons). The objectives of this work are (1) to explore whether the freshwater wetland
acts as a potential sink or source of atmospheric VOCs and the effects of litter decomposition,
(2) to investigate the seasonal pattern of VOC fluxes and (3) to analyze potential influencing



factors modulating VOC fluxes during the litter decomposition. The insights gained from this
study can serve as a reference for models or emission inventories to estimate global BVOC
emissions and enhance our understanding of the role of freshwater wetlands in reactive VOC
budget.
**2. Methodology**
**2.1 Study area**
In-situ field experiments were carried out at Kuihu (KH, 30.32°-31.57° N, 117.67°-118.73° E),
a typical freshwater wetland in southeastern Anhui Province (Fig. S1), about 20 km from the
city. The total area of KH is 5.05 km$^2$ and its wetland area is 4.46 km$^2$
(https://www.wuhu.gov.cn/openness/public/6596211/15236661.html). Based on previous
statistical information, vegetation in local wetland is predominated by *Phragmites australis*
(the statistical document is available on request by writing to corresponding author). Here, we
thus used *Phragmites australis* litter as typical case to explore the VOC flux variations during
the litter decaying in the freshwater wetland.
**2.2 Experimental setup**
The aim of this study was to investigate that whether the freshwater wetland acts as the sink
or source of VOCs and the effects of litter decomposition, thus we selected sunny days and
avoid rainy/cloudy days, which could underestimate these VOC emissions (Li et al., 2023), to
conduct field measurements. The sampling time of 11 campaigns occurred at around
10:00-14:00. As showed in Fig. 1 (a), nine 1.1m × 1.1m × 1.1m (length × width × height)
stainless steel cuboidal boxes without the cover were installed in the flooded area of
freshwater wetland. The distance between the adjacent boxes was set as one meter. During the
experiments, three treatments, namely A, B and C, were designed and each of treatment
consisted of three parallel groups. For A, B and C, no plant litter addition, 1.4 kg litter
addition and 2.8 kg litter addition, respectively, were treated. Nine nylon mesh bags were
attached to each box, as shown in Fig. 1 (b). Each mesh bag in A, B and C contained 0 kg, ~
0.156 kg and ~ 0.311 kg litter. The *Phragmites australis* litters were collected directly from
Kuihu wetland and weighted by an electronic balance (HY-809B, ZHIZUN, China). Except
the litter mass, all settings of the three treatments were kept as constant as possible in our



experiments. Moreover, to avoid significant interference on the wetland ecosystem, only
*Phragmites australis* above the roots were cleared up and the roots remained in nine boxes as
showed in Fig. 1 (b).
**2.3 Flux measurements**
In situ static-chamber measurements for VOC fluxes were carried out in this study and more
detailed description about this method can be found in our previous studies (Wang et al., 2015;
Liu et al., 2021). The chamber (35cm × 35cm × 15cm) made of stainless-steel frames covered
with Teflon film was placed inside the cuboid box. The bottom of chamber was installed with
a urethane foam board, which allowed chamber to float on the water. Air samples inside the
chamber were collected by pre-evacuated 3.2-L silonite-treated stainless steel canisters
(Entech Instruments Inc., Simi Valley, CA, USA) at 0 min, 10min and 30min. A Teflon tube
extended into the center point of the chamber was used to draw air into the pre-evacuated
canister.
The calculation of net VOC fluxes was based on following equation:
$$Flux = \frac{dC}{dt} \times \frac{V_c}{A_c}$$
where $\frac{dC}{dt}$ is the linear regression slope of the chamber headspace VOC concentration (μg m$^{-3}$)
versus time (min). $V_c$ (m$^3$) is the volume of the chamber. $A_c$ (m$^2$) is the base area of the
chamber that floated on the water. In fact, we found that VOC concentration in chamber
headspace has already leveled off at 30 min. Thus, here the first two points, which could
capture the initial fluxes, were used to calculate VOC fluxes (Zhang et al., 2021).
**2.4 Laboratory analysis**
*VOCs*
The air samples were analyzed by a model 7200 pre-concentrator (Entech Instruments,
California, USA) coupled to a gas chromatography and mass selective detector (GC/MSD,
Shimadzu Corp, Tokyo, Japan). The analysis steps were described in detail elsewhere (Liu et
al., 2021; Wang et al., 2023). Briefly, air samples were firstly drawn into a liquid-nitrogen
cryogenic trap at -160 °C to be trapped and concentrated, then transferred to a secondary trap
at -40 °C by pure helium. After these two steps, the $H_2O$ and $CO_2$ were mostly removed.



Subsequently, secondary trap was heated, and the VOCs were brought by helium to a third
cry-focus trap at -170 °C. Once air samples were focused, the third trap was heated rapidly to
be transferred into GC/MS system for further VOC analysis. The mixture was separated by a
DB-1 capillary column (60m × 0.32mm × 1.0 μm, Agilent Technologies, USA) with helium
as carrier gas. The initial oven temperature was set as 10 °C, kept 3 min, then increased to
120 °C at 5 °C min$^{-1}$, and then changed to 10 °C min$^{-1}$, finally reached at 250, held for 2 min.
MSD was operated in SCAN model with electron impacting (EI, 70 eV) as the ionization
method here. The Text S1 in supplement provided the information about quality assurance
and quality control (QA/QC).
***CH$_4$ and CO$_2$***
CH$_4$ and CO$_2$ were measured from canister air samples by gas chromatography (GC-2014,
Shimadzu, Kyoto, Japan) equipped with a flame ionization detector (FID). CH$_4$ was directly
detected by FID. While CO$_2$ firstly was converted to CH$_4$ by flushed with H$_2$ through hot
nickel power catalyst, then detected by FID (Miao et al., 2022).
**3. Results and discussion**
**3.1 Overview of VOC fluxes**
Throughout the entire in-situ observation, 62 VOCs including 19 alkanes, 11 alkenes, 5
aromatics, 15 OVOCs, 9 halocarbons and 3 VOSCs were detected and quantified in this study
(Table S1). From the Table 1, regarding total net fluxes, three treatments all presented positive
VOC fluxes, suggesting that the freshwater wetland could be a potential source of
atmospheric VOCs. Obviously, as showed in Table 1, litter decomposition enhanced the VOC
emissions from freshwater wetland. The average values of total net VOC fluxes measured in
B and C were 5.93±3.13 μg m$^{-2}$ h$^{-1}$ and 8.30±4.00 μg m$^{-2}$ h$^{-1}$ (average ± standard error),
respectively, ~ 2.04 times and ~ 2.86 times that of A (2.90±2.74 μg m$^{-2}$ h$^{-1}$). The effects of
litter decomposition varied depending on VOC groups. For example, net fluxes of alkanes and
alkenes measured in C were unexpectedly lower than in B. Moreover, the net flux of
halocarbon measured in B was similar to that in C. These results reflected that besides the
litter mass, other factors such as microbe activities (Lorah et al., 1999; Leff et al., 2008; Gray
et al., 2014; Jiao et al., 2018; Svebdsen et al., 2018), which were discussed in detail in *Section*
*3.3*, might also affect the VOC fluxes measured in our experiments.



From the Table 1, both halocarbons and OVOCs showed net uptake into wetland in A. When
inputting the litters, halocarbons shifted from net uptake into wetland to net positive efflux
out of wetland. OVOC fluxes in A and B were both negative but showed a positive value in C,
indicating that litter decomposition could be one of the crucial factors determining whether
the freshwater wetland was a net source or a net sink of atmospheric OVOCs. As showed in
Fig. 2, litter decomposition apparently affected the compositions of VOCs released from
freshwater wetland. Notably, aromatic, a typical class of anthropogenic air pollutant, was the
dominant VOC group released from freshwater wetland, accounting for 40.6%, 40.4% and
46.3% of total VOC emissions in A, B and C, respectively (Fig. 2). Recently, several studies
gave the evidence of biogenic aromatic emission from living plants, straw, cyanobacteria and
ocean phytoplankton (White et al., 2009; Rocco et al., 2021; Liu et al., 2021; Wohl et al.,
2023; Wu et al., 2023).
As Fig. S2 shows, during the year-round observation, the fluxes of VOC-driven carbon ($C_{VOCs}$)
in freshwater wetland averaged 2.47±1.84, 4.69±2.11 and 6.36±2.78 $\mu gC\ m^{-2}\ h^{-1}$ in A, B and
C, respectively. The $C_{VOCs}$ accounted for < 0.05% of the carbon driven from $CO_2$ and $CH_4$. In
comparison to $CO_2$ and $CH_4$, VOC emission was not a significant pathway for wetland-carbon
transporting into atmosphere. However, given their relatively higher reactivities, these VOCs
preferentially reacted with OH radicals once being released into atmosphere from freshwater
wetland. This would prolong the atmospheric lifetime of $CH_4$, thus accelerating global
warming. While global warming would in turn lead to greater release of VOC from the
freshwater wetland as temperature was linearly correlated with VOC emission according to
the discussion in *Section 3.2*. It is clear from Fig. S3 that these above-mentioned processes
would develop a vicious circle.
Previous studies also revealed that the wetland environment was a source of atmospheric
VOCs, but what these studies mostly investigated were living plants and focused on the fens
(Hellén et al., 2020; Seco et al., 2020; Vettikkat et al., 2023). As showed in Table S2, the net
VOC fluxes measured in this study, representing the freshwater wetland, were lower than
those obtained in fens. It could be realized that VOC emissions from fens were dominated by
isoprene, with fluxes ranging from 2.5 $\mu g\ m^{-2}\ h^{-1}$ to 4773.6 $\mu g\ m^{-2}\ h^{-1}$. These were quite
different from the results uncovered by our observation. The difference could be attributed to



the variety of sampling, analysis methods and detected VOC species among the studies on the
one hand and to the wetland types on the other hand. In the present work, our experiments
were carried out in flooded areas of the freshwater wetland and focused on dead litter
decomposition, while earlier studies focused on relatively drier environments and living
plants. Baggesen et al. (2022) pointed out that flooded condition would result in a negative
net VOC flux and even overruled other effects. In addition, the GC/MSD technique used here
failed to fully characterize higher molecular weight species, such as monoterpene, and lower
molecular weight species, such as methanol, which have been reported to be released in
significant amounts during the decomposition of litter (Gray et al., 2010; Faiola et al., 2014).
**3.2 Seasonal patterns**
As shown in Fig. S4, the net VOC fluxes measured in three treatments all demonstrated
significant seasonal variations. VOC uptake from air to wetland generally occurred in cold
winter while net VOC emissions were found in warm and hot seasons. The highest VOC
emission was found in hot summer, followed by spring and fall. The overall seasonal pattern
of VOC fluxes observed here was similar with the previous studies conducted in wetland
environments (Hardacre et al., 2013; Seco et al., 2020).
In spring, as Fig. 3 shows, positive net VOC fluxes were showed in three treatments during
the samplings conducted in March and April. However, the negative net VOC flux was found
in A and the positive net VOC fluxes of B and C were also significantly lower observed in
May than those measured in prior two samplings, which could be attributed to the sharp drop
in temperature (-6.6 °C). As shown in Fig. 3, VOC emission enhanced by litter decomposition
were much remarkable in hot summer. Considering that temperature was the key factor
driving the seasonal patterns of VOC fluxes measured during the experiments, we further
analyzed the relationships of net VOC fluxes with ambient temperature. As shown in Fig. 4,
VOC fluxes presented significant positive correlations with temperature ($p < 0.05$) in three
treatments. According to the linear fitting equations showed in Fig. 4, VOC uptake from air
into wetland generally occurred at lower temperature (< 10 °C), and similar VOC uptake was
found between B and C, both presenting less VOC uptake than A. In fact, lower temperature
would limit microbial activities and was an unfavorable condition for decaying litters



(Greenberg et al., 2012; Gray et al., 2014). Thus, higher VOC uptake measured in A could be
mainly caused by abiotic process (Gray et al., 2014). Compared to A, the litter added in B and
C covered the water surface, preventing VOC uptake from ambient air into wetland.
Moreover, slow decaying of litter in B and C possibly also occurred even under low
temperature. This process might release some VOCs into ambient air and offset part of VOC
uptake, thus resulting in B and C lower net negative VOC fluxes than that showed in A. As
displayed in Fig. 4, the slopes of linear fitting equation were 0.72 and 0.81 for B and C,
respectively, higher than that of A (0.63). This further clarified that litter decomposition would
stimulate VOC emission from freshwater wetland and suggested that the positive effect of
litter decomposition was amplified by higher temperature.
In fall, it was worth noting that the negative VOC flux was measured in September (25.9 °C),
while positive VOC flux was found in October despite of its relatively lower temperature
(16.1 °C). This reflected that in addition to ambient temperature, the uptake or release of
VOCs could be influenced to some extent by other factors, thus seasonal variations of the net
VOC fluxes could be the combined results of these complicated factors.
**3.3 The flux variations of specific VOC groups**
As mentioned in *Section 3.1*, uptake or release of VOCs varied to their chemical groups. Here,
the measured fluxes of different VOC groups were analyzed in detail to explore the potential
factors determining their release or uptake.
**3.3.1 Alkanes**
Overall, freshwater wetland was the potential source of atmospheric alkane, as positive net
fluxes were measured in three treatments during the year-round observation (Table 1). For A,
B and C, alkanes accounted for 23.3%, 20.4% and 13.8% of total VOC emissions,
respectively (Fig. 2), which were comparable to that previously reported in decomposition of
tree leaf (~ 20%) (Svendsen et al., 2018; Viros et al., 2020). The emission percentage of
alkane showed a decreasing trend with litter addition, suggesting that alkane might not be the
dominant VOC composition emitted during litter decomposition in freshwater wetland. In
general, higher temperature would stimulate microbe activities and accelerate litter decaying,
further leading to more VOC release (Isidorov et al., 2002; Jiao et al., 2018; Svendsen et al.,
2018). However, from the Fig. 5, it was interesting that B and C presented higher alkane





emissions than A during the period with relatively lower temperature, while nearly equal
amounts of alkane emissions were observed in three treatments in hot July. This result
suggested that higher alkane emission in July might not be mainly resulting from litter
decomposition. Previous studies have revealed that alkanes can be stored in the upper cuticle
layer of vascular species (Bondada et al., 1996; Barik et al., 2004). Thus, during the initial
period of litter decomposition (lower temperature), higher emission fluxes of alkane measured
in B and C might be mainly due to the material composition of the litter itself rather than
resulting from the litter decomposition. Notably, in subsequent experiments such as those
conducted in March, June and July, relatively higher alkane emission was measured in A,
even comparable to that in B and C. This suggested that litter decomposition might not be
main source of alkanes and that other alkane-emitted sources might also exist in freshwater
wetland. In fact, earlier studies have found that root-emitted VOCs could act as chemical
signals belowground and be transported from the source (Delory et al., 2016). Root exudation
in the rhizosphere contained numerous chemicals like lower-molecular-weight organic acids,
which were reported to be the precursors of biogenic alkanes (Maffei, 1994; Viros et al.,
2020). Considering that the roots of *Phragmites australis* were not removed in our
experiments (Fig. 1), root emission might be an alkane source, which needed more
investigations in future to explore and confirm this.
**3.3.2 Alkenes**
From the Table 1, freshwater wetland was the source of atmospheric alkenes and litter
decomposition positively stimulated alkene release from the wetland. In contrast to alkanes,
relatively lower alkene emission, as displayed in Fig. 5, was observed in the initial stage of
litter decomposition, and meanwhile the effect of litter decomposition was not significant.
However, in hot July, alkene emission was much higher in B and C than that in A, suggesting
that alkene emission arising from litter decomposition was reinforced by higher temperature.
It was worth noting that similar alkene emissions were found between B and C in hot July,
suggesting that besides litter mass, alkene flux was also subject to other factors such as
microbial activities (Svendsen et al., 2018). Additionally, alkene flux measured in A was
similar from March to July expect April when the temperature dropped sharply. It has been
previously reported that alkenes are the fatty acid derivatives in natural environments (Maffei,





1994; Viros et al., 2020), and thus root exudates and wetland microbes which were likely to
produce fatty acid might also contribute to the alkenes.
**3.3.3 Aromatics**
Traditionally, aromatic hydrocarbons (AHs) have been regarded as anthropogenic air
pollutants (Barletta et al., 2005; Kansal et al., 2009; Lamarque et al., 2010). However,
biogenic emission of AHs has attracted attention from atmospheric science community and
has been reported in recent studies (White et al., 2009; Misztal et al., 2015). During the
experiments, we also detected six AHs and found that they were even the largest VOC species
released from the freshwater wetland. From the Fig. 2, the percentage of AHs increased with
litter addition. Moreover, throughout the entire observation, AH fluxes measured in B and C
were higher than that in A (Table 1). These results reflected that litter decomposition could
enhance AH release from the freshwater wetland. In terms of individual AH species, as
showed in Fig. S5, we found that m/p-xylene was the largest AH species in A, while toluene
ranked No.1 contributor to total AH emission in B and C. This suggested that litter
decomposition could release more toluene than other detected AH species. Biogenic AH
emissions, particularly for toluene, have been observed in terrestrial, aquatic and marine
ecosystems (Liu et al., 2021; Rocco et al., 2021; Wohl et al., 2023; Wu et al., 2023). Previous
studies reported that AHs were emitted during the decomposition of tree leaf, but only
accounted for < 10% of total VOC emission (Faiola et al., 2014), quite different from our
results (> 40%, Fig. 2). Two explanations could be that, on the one hand, AH emission could
vary depending on litter types (Gray et al., 2010) and, on the other hand, that previous studies
investigated litter decomposition in aerobic environments, while our experiments were
conducted in flooded, relatively low-oxygen circumstances. A recent study measured VOC
emission during straw decomposition and found that AH emission could reach over 3 times
higher under flooded conditions than under non-flooded conditions (Wu et al., 2023).
Additionally, earlier studies revealed that biogenic toluene is preferentially produced under
anaerobic conditions (Jüttner and Henatsch, 1986; Mcrowiec et al., 2005; Moe et al., 2018).
Therefore, in our experiments, the low-oxygen environment could be favorable for biogenic
AH production.
**3.3.4 Halocarbons**

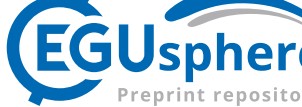

Overall, net halocarbon emission was relatively lower, accounting for 7.2% and 5.9% of total
VOC emission in B and C, respectively (Table 1). A negative flux was measured in A,
suggesting that the freshwater wetland could be the net sink of halocarbon. Undoubtedly, litter
decomposition enhanced the halocarbon emission, especially in hot July and August, but the
emission did not fully follow the seasonal patterns. As showed in Fig. 5, both B and C still
presented significant halocarbon emissions in October, and even in November, when the
ambient temperature was below 10 °C, halocarbon emission in C was comparable to that
measured in hot July. These results suggested that halocarbon emission was not majorly
controlled by temperature. Natural emissions of halocarbons have been previously reported in
marine (Philip et al., 1985), soil (Albers et al., 2017) and forest wetland (Jiao et al., 2018).
Typical halocarbon species emitted from natural source such as methyl chloride ($CH_3Cl$) and
chloroform ($CHCl_3$) were also detected in our experiments. Both $CH_3Cl$ and $CHCl_3$ have been
reported to be formed through biotic (enzymatic reactions) (Rhew et al., 2003) and abiotic
processes (nonenzymatic Fenton-like reactions) (Huber et al., 2009). Jiao et al. (2018)
revealed that abiotic factor dominated the $CHCl_3$ emission in forest wetland, while microbial
activities controlled the $CHCl_3$ production in Antarctica tundra soils (Zhang et al., 2021).
These conflicting results reported by earlier studies reflected that biogeochemical process of
natural halocarbon were very complicated and varied to natural environments. During our
experiments, the $CH_3Cl$ fluxes in A, B and C were -0.08±0.12, 0.27±0.12 and 0.39±0.12 µg
$m^{-2}h^{-1}$, respectively, but $CHCl_3$ was undetectable in many measurements. Both the net total
halocarbon fluxes and $CH_3Cl$ did not show significant correlations with ambient temperature
(Table S3), which was in line with the consequence reported by Jiao et al. (2018). This
suggested that halocarbon emission from freshwater wetland might be predominantly
controlled by abiotic process rather than temperature-dependent microbial activities (Gray et
al., 2014).
**3.3.5 Oxygenated volatile organic compounds**
Unlike other VOCs, OVOCs presented uptake in most flux measurements conducted in the
freshwater wetland (Fig. 5). This was quite different from previous studies on the
decomposition of tree leaf litter, which released large amounts of OVOCs. For example, Gray
et al. (2010) reported that OVOCs accounted for over 99% of total VOC emissions during



plant litter decomposition. However, the litter decomposition investigated in these previous
studies generally took place in dry terrestrial environments or in laboratory incubations, while
our experiments were conducted in freshwater wetland under flooded conditions. As we know,
most OVOCs were polar organic molecules including alcohols, carbonyls and organic acids,
which are water soluble (Baggesen et al., 2022). Bourtsoukidis et al. (2018) reported that soil
could shift between methanol release when dry and methanol uptake when wet. OVOC
deposition was also found in early morning due to dew (Schallhart et al., 2016). Moreover,
previous studies reported OVOC uptake in ocean and lake environments (Seco et al., 2020;
Liu et al., 2021). Even in hot July, OVOCs such as acetone and acetaldehyde showed negative
net fluxes in the field observation of freshwater (Seco et al., 2020). Thus, even if OVOCs
could be produced during our field experiments, they were likely to dissolve in the water.
From the Fig. 5, three treatments all displayed OVOC uptake at initial stage (January) of litter
decomposition. On January 9th, the highest and lowest OVOC uptake was observed in A and
C, respectively, while completely opposite results were observed on January 21st. As known,
OVOCs can be consumed by microbes (Männistö et al., 2023). For B and C, litter
decomposition over time could provide a carbon source for microbes (Albers et al., 2018) and
increase their amounts and activities, further enhancing OVOC consumption by microbes.
This could result in increasing uptake of airborne OVOCs into freshwater wetland. It was
worth noting that in hot July, three treatments all presented positive flux values of OVOCs,
indicating that OVOCs actually could be produced in freshwater wetland. However, this
production might be hidden by its sink such as dissolution in water, microbial uptake and
abiotic deposition due to the concentration gradient between wetland and atmosphere
(Niinemets et al., 2014; Alber et al., 2017; Rinnan et al., 2020). Further, litter decomposition
enhanced OVOC emission in hot July. Although higher temperature could accelerate
microbial decomposition of litter and release more OVOCs, considering their water solubility,
higher OVOC emission fluxes were likely the outcome of strong evaporation, a typical abiotic
process (Leff and Fierer, 2008), in hot July.
**3.3.6 VOSCs**
In this study, we detected three VOSC species including dimethyl sulfide (DMS), dimethyl
disulfide (DMDS) and dimethyl trisulfide (DMTS), which were widely measured in aquatic



ecosystems (Liu et al., 2021). During the entire observed period, VOSC release from
freshwater wetland were found in three treatments (Table 1) and DMS dominated the VOSC
emission, accounting for 78.8±2.9% (Fig. 6). Overall, litter addition enhanced VOSC
emissions from the freshwater wetland. However, as Fig. 5 shows, an abnormal phenomenon
was observed in August, which presented highest VOSC emission in B, followed by A and C.
As previously reported, VOSCs were generated from microbial metabolism of methionine and
cysteine (Achyuthan et al., 2017). Thus, all of these could produce methionine and cysteine
such as organics dissolved in water were probably responsible for higher VOSC emission in B
in August. As Fig. 5 shows, significant VOSC emission was still found in October (T=16.1°C)
solely under the B and C, similar to that of AHs, halocarbons and OVOCs. This could be
attributed to the consequence of litter decomposition by microbes.
**4. Conclusions and limitations**
In this study, in situ flux measurements of VOCs were conducted in a freshwater wetland
during year-round litter decomposition. Three treatments including A (no litter addition), B
(1.4 kg litter) and C (2.8 kg litter) were designed to investigate the impacts of litter
decomposition on VOC fluxes measured during the field experiments. As a result, the net
TVOC fluxes measured in A, B and C were 2.90±2.74, 5.93±3.13 and 8.30±4.00 μg m$^{-2}$ h$^{-1}$,
respectively, suggesting that the freshwater wetland was a potential source of atmospheric
VOCs and litter decomposition could enhance its net VOC release. Although revealed as the
source of VOCs in terms of net total flux, uptake or release of VOCs observed in freshwater
wetland varied depending on the specific VOC chemical groups. NMHCs and VOSCs both
showed positive net fluxes in three treatments. Halocarbons presented a negative net flux in A
and shifted to positive fluxes in B and C. Due to their water solubility, OVOCs measured here
mostly presented uptake in the wetland, and their fluxes were found to be negative in A and B,
but turn to be positive in C. This reflected that litter decomposition could result in freshwater
wetland becoming a source of atmospheric OVOCs. In line with previous studies on BVOC
emissions, net TVOC flux showed significant seasonal patterns. In three treatments, the
highest positive fluxes were observed in hot summer, followed by spring and fall, while the
negative net VOC fluxes were generally found in cold winter. Moreover, VOC emission
increasement resulting from litter decomposition exhibited monotonically increasing with



litter mass in the seasons except for fall, of which the highest flux presented in B, followed by
C and A. In three treatments, net VOC fluxes were significantly correlated with ambient
temperature ($p < 0.05$). Based on detailed analysis for flux variations of different VOC groups,
biotic and abiotic processes driven by temperature co-modulated VOC fluxes measured in
freshwater wetland.
Overall, our study provides some new insights into the role of freshwater wetland in reactive
VOC budget. The results gained from this work could be indicative for current models
simulating BVOC emission as well as for biogeochemical process of reactive VOCs. Of
course, there are some limitations in this work that should be pointed out. First, our
measurements were all performed in daytime using off-line canister sampling. In future,
online observations considering both daytime and nighttime VOC fluxes are needed to
elucidate the role of the freshwater wetland in VOC budget more comprehensively. In
addition, we did not directly link the measured VOC fluxes to specific microbes in this study.
The next step should focus on the key microbes related to the release or uptake of VOCs and
uncover more detailed biogeochemical mechanisms of VOCs in freshwater wetlands.



**Data availability**

All raw data can be provided by the corresponding author upon request.

**Financial support**

This work has been funded by the National Natural Science Foundation of China (42207128 and 41273095), Natural Science Foundation of Anhui Province (2008085MD111), Key Research Projects of Natural Science in Colleges and Universities of Anhui Province (KJ2021A0091).

**Competing interests**

The contact author has declared that none of the authors has any competing interests.

**Author contributions**

H.F., S.M., F.Z., J.Z., Z.Y., H.X., Y.H. conducted the field experiments and collected the samples. H.F., S.M., Q.J., J.Z. analyzed the samples. H.F., S.M., T.W., F.Z., J.Z. designed the experiments. H.F., T.W. provided the funding supports. H.F. wrote the paper. T.W., and X.W. revied the paper.



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



**Table 1.** The net fluxes (mean ± standard error) of different VOC groups under A, B and C
treatments (unit: μg m$^{-2}$ h$^{-1}$) based on 11 campaigns during one-year litter decomposition.
Each treatment consists of three parallel groups. The number of measured compounds is
given in parentheses.

| | A | B | C |
| --- | --- | --- | --- |
| | (No litter addition) | (1.4 kg litters) | (2.8 kg litters) |
| Alkanes (19) | 0.76±0.35 | 1.45±0.48 | 1.20±0.64 |
| Alkenes (11) | 0.55±0.29 | 1.40±0.37 | 1.28±0.41 |
| Aromatics (5) | 1.33±0.68 | 2.86±0.84 | 4.02±1.04 |
| Halocarbons (15) | -0.09±0.18 | 0.51±0.17 | 0.43±0.20 |
| OVOCs (9) | -0.02±1.65 | -0.07±1.73 | 0.64±2.48 |
| VOSCs (3) | 0.64±0.40 | 0.88±0.50 | 1.03±0.22 |
| Σ | 2.90±2.74 | 5.93±3.13 | 8.30±4.00 |




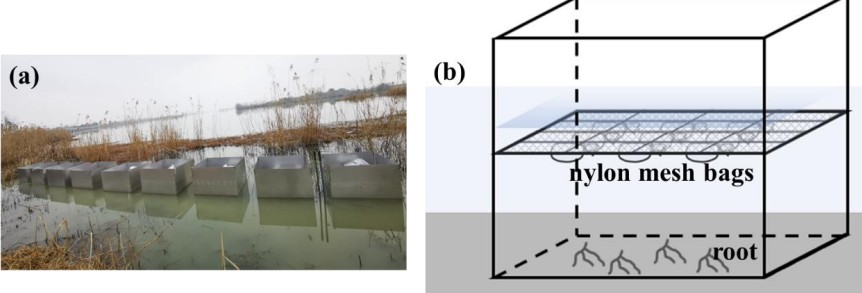


**Figure 1.** Nine stainless steel cuboidal boxes without the cover in freshwater wetland (a). The
schematic diagram of cuboidal box was displayed in (b). Each treatment, namely A, B and C,
consists of three parallel groups.



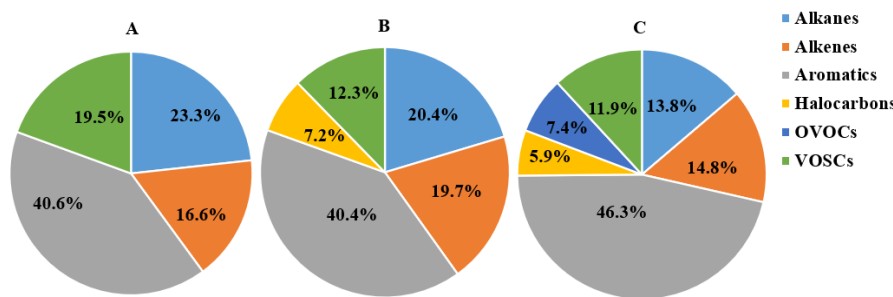

**Figure. 2** Chemical compositions (mass percentage) of net VOC emissions under A, B and C treatments.



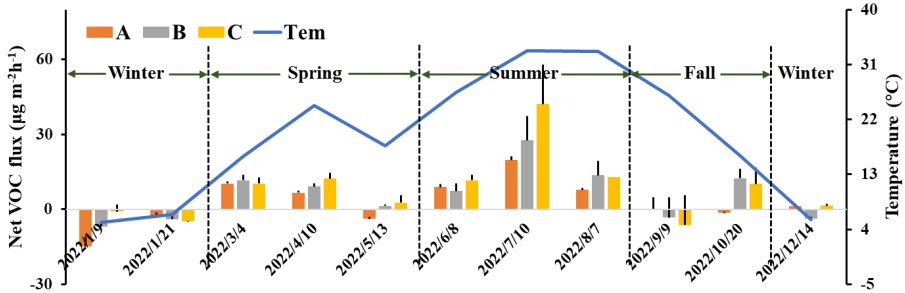

**Figure 3.** Seasonal variations of net VOC fluxes under three treatments. The error bar (if more than two values) represents standard error of three parallel samples.



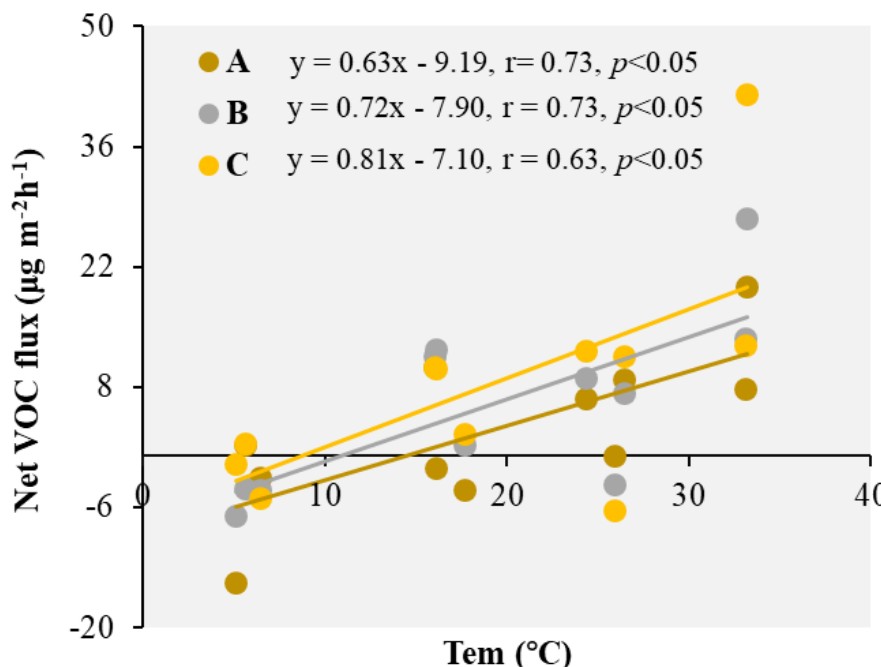

**Figure 4.** The relationships of net VOC fluxes with ambient air temperature (Tem) during one-year litter decomposition.

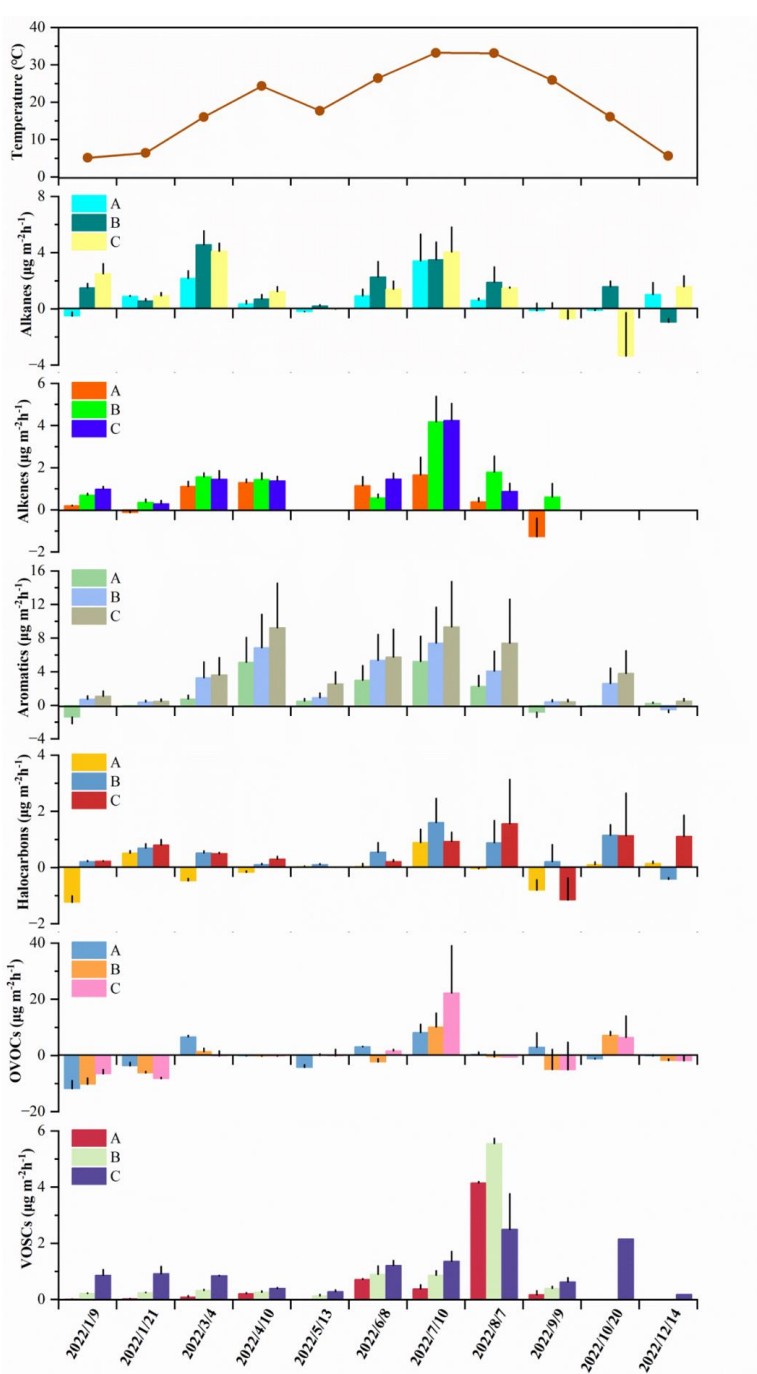

**Figure 5.** The time series of air temperature and fluxes of different VOC groups. The error bar (if more than two values) represents standard error of three parallel samples.