# Peer review of "Litter decomposition enhances volatile organic compound emission from a freshwater wetland: insights from year-round in situ field experiments"

_EGUsphere, 2023_

## Referee Comment (RC2)

Title: Litter decomposition enhances volatile organic compound emission from a freshwater wetland: insights from year-round in situ field experiments
Author(s): Hua Fang et al.
MS No.: egusphere-2023-2998

The authors report VOC fluxes from the decaying leaf litter of a freshwater wetland plant, Phragmites australis. The reported fluxes include an impressive list of 62 VOCs: 5 aromatics, 15 OVOCs, 30 NMHCs, 9 HVOCs, 3 VOSCs.  Many of these compounds are difficult to quantify well, thus requiring proof of careful methodology.  The proof, unfortunately, is not provided, leading to doubts about data quality. The problem is compounded by the use of incomplete references. The methods section cites two papers (Wang et al, 2023 and Liu et al. 2021) to broadly explain 'analysis steps'.  The cited studies employ slightly different methods and themselves cite a range of earlier papers for broad methodological statements.  It becomes a nearly impossible task to follow the actual methods of VOC quantification. Specific papers should be cited for specific methods, and only if they truly align. For example, Wang et al., 2023, uses a different instrument altogether (GC-MSD-FID from Agilent vs. a GC-MSD from Shimadzu), so it should be made evident why this paper is cited.

Below, I compile a checklist for methodology that should be included for field measurements of VOC fluxes.  While several items are addressed in this study, other critical aspects are missing.

| SAMPLE STORAGE | |
|---|---|
| Canister blank | Yes.  Filled with N2, evacuated, tested for contamination. Described in S1 |
| Canister stability | Stability of compounds was not tested for the compounds of interest, even though they have different degrees of stability in canisters. Compounds with poor stability or even production within canisters from other compounds (e.g., VOSCs) need to be checked. |
| | |
| FIELD SAMPLING | |
| Flux chamber blank (sampling equipment only) | Chamber blank/control experiments using empty chambers were not conducted, which is concerning because of the use of foam board to make the chamber float.  Separate issue: chambers are described as 1.1m x 1.1m x 1.1m (Line124), but it appears that another smaller chamber (35x35x15cm) is then placed inside the larger box (Line 140), is that correct? |
| Leakage / sample replacement | A correction is needed to account for the removal of 3.2 L samples taken from an 18 L chamber.  This is not a large correction, given that the first sample is taken at the beginning when outside air = inside air, but it should be considered (if using 3 points instead of 2). Leakage/sample replacement may be responsible for the non-linearity of concentration change that was observed. |
| Replication | Yes.  3 times per treatment of leaf litter (Line 126-128). |
| | |

| INSTRUMENT | |
|---|---|
| Instrument blanks | Yes. N2-filled canisters analyzed.  Described in S1 |
| Hardware and chromatography details | Yes. Entech pre-concentrator with Shimadzu GC-MSD. Temperature programming.  Note that running in SCAN mode is less precise than SIM (Lines 156-170) |
| Method of Peak ID | Yes. By retention time and m/z matching standards. |
| Instrument precision | Not reported.  Precisions are needed for each compound. |
| Duplicate runs | Not reported. |
| Detection limit | Not reported. |
| Use of Gas standards | Diluted PAMS and TO-15 standard mixtures are cited.  DMS and DMDS standards were used but not described.  For VOCs without standards, they were "semi-quantified based on the species whose RTs were close to them or based on the species which have a similar chemical structure to them."  This is not an appropriate method, as there is no reason to believe the instrument sensitivity or ion fragmentation will be the same.  These compounds must be removed from any quantitative analysis. |
| Gas standard traceability | Missing.  Need to add source of standard and standard traceability (if available) |
| Calibration curves | Calibration curves were run at: 0.5, 1, 5, 15, 30 ppb for PAMS and TO-15. 0.1, 0.5, 1, 2, 4 ppb for sulfur. |
| Cal Curve over range of concentrations | May not be valid, as VOC concentrations / ranges were not reported.  Calibration curves might not be appropriate for the range of concentrations observed.  If concentrations <0.5 ppb (or <100 ppt for VOSCs), then the calibration curve is a one point calibration. |
| | |
| **FLUX CALCULATIONS** | |
| Curve fitting procedures | "VOC concentrations in chamber headspace had already leveled off at 30 in. Thus, here the first two points, which could capture the initial fluxes, were used to calculate VOC fluxes (Zhang et al., 2021)."  What is meant by 'leveled off', and do all chambers show this trend?  Zhang et al. only used 2 point flux calculations for one compound and only in the <10% of cases when r2<0.75.  Here, the 2-point method appears to be used for all VOCs. Is the system leaky or is there a dilution effect from withdrawing large samples (see above)? |
| Flux detection limits | Given the precision of each compound, what are their flux detection limits?  When is a flux determined to be not significantly different from zero?  How are these observations treated? |

In summary, numerous revisions are needed to address the methodological omissions and to improve the quality of flux data. Measuring 62 VOC fluxes is impressive, but how many of these fluxes are real?  I suspect several of the measurements have larger uncertainties than presented and/or are below detection levels. Also, aggregating fluxes by VOC type can summarize a large set of results, but this process obscures what is happening with specific

compounds. By aggregating by mass, heavier compounds can dominate the overall picture. Please provide more information about the measurements that are defensible, add caveats where needed, and exclude results that are unreliable.  Only then can the reader be fully able to assess the scientific contributions of the work.

---

## Author Comment (AC1)

**Reviewer #1**

Fang et al., in their article "Litter decomposition enhances volatile organic compound emissions from a freshwater wetland: insights from a year-round in situ field experiment", present results from year-round VOC fluxes in a freshwater wetland located in southeastern Anhui Province in China for three different treatments: no litter addition, 1.4kg litter, and 2.8kg litter.

The manuscript is clear and the arguments easy to follow. The presentation of the results is clear, and they are discussed appropriately. I only found minor issues, which I trust the authors will be able to address. Therefore, I recommend accepting the manuscript with minor corrections.

**Reply:** Thank you for the helpful comments and providing us the opportunity to revise the manuscript. We have carefully addressed the comments in point-by-point form as shown below.

Minor comments:

- ll. 128-131: An important element that is missing from the manuscript is an indication of naturally occurring litter. How much is it in treatment A? Are the authors able to give an estimate?

**Reply:** Thanks for the comments. In fact, in treatment A, the *Phragmites Australis* above the roots were removed and no plant litter was added. This treatment was used to compare the wetland-air exchange of VOCs under no plant litter decomposition.

Moreover, as for naturally occurring litter, you pointed out an interesting question, which was also in our previous consideration. However, we finally have not included the estimations and discussion on this part in our manuscript for two reasons as below: First, in addition to *Phragmites Australis*, there are several other plants that live in the wetland. These plants have different growth cycles and could be also affected by the current global warming trend, which poses a challenge for estimating naturally occurring litter. It is therefore unfortunate that we are currently unable to provide such an estimate.

Second, the objectives of this study are to investigate the wetland-air exchange of VOCs and the impacts of plant litter decomposition. Using 11 in-situ field experiments in the wetland, we obtained the results that allow us to achieve above aims. Estimating naturally occurring litter is quite interesting and meaningful work but is obviously outside the scope of the present study. Anyway, you have made a valuable suggestion that we will consider it in our next work.

- ll. 134-136: Could the authors elaborate on when the Phragmites Australis above the roots were cleared in relation to the first samples being taken? In addition, could the authors briefly mention why roots were/could not be cleared up?

**Reply:** Thank you for the comments.

1) The *Phragmites Australis* above the roots were removed on December 21, 2021, eighteen days before the first formal sampling on January 9, 2022.

2) When we conducted the in-situ field experiments, the roots of *Phragmites Australis* were not removed. On the one hand, as mentioned in the original manuscript, we have, as far as possible, not made much significant intervention in the wetland ecosystem. More importantly, this is more likely to follow the process of plants' natural apoptosis, during which the roots remain in the wetland soil.

In the revised manuscript, we also added above information as follows:

"*only Phragmites australis above the roots were cleared up on December 21, 2021, eighteen days before the first formal sampling (January 9, 2022). As showed in Fig. 1 (b), the roots were kept in the nine boxes and this is more likely to follow the process of plants' natural apoptosis, during which the roots remain in the wetland soil.*" (Line 137-141)

- ll. 213-214 (and Fig. S3): I wonder if Fig. S3 is necessary and if the authors should not instead refer to the literature to back their statement about this positive feedback loop. In addition, in Fig. S3 'arise' is the only intransitive verb and should be replace by 'increase', for instance, to be consistent with the two other (transitive) verbs.

**Reply:** Thank you. As reviewer suggested, we have deleted Fig. S3 in supplement.

- ll. 227-230: The authors only state that the 'technique used here failed to fully characterize higher molecular weight species, such as monoterpenes and lower molecular weight species, such as methanol'. Could the authors be more specific in relation to the limitations of the method? Does it have to do with the sampling? Also,

do the authors mean 'higher molecular weight species, such as sesquiterpenes' as they include at least two monoterpenes in their results (Table S1).

**Reply:** Thank you for carefully reviewing our manuscript. As you pointed out, the higher molecular weight species were sesquiterpenes rather than monoterpenes. In the updated version, we have modified this. Moreover, the method limitations for higher molecular weight species such as sesquiterpenes and lower molecular weight species such as methanol were described as below:

"*Previous studies reported the release of significant amounts of sesquiterpenes and methanol during the decomposition of litter (Gray et al., 2010; Faiola et al., 2014). However, due to wall loss, the canister sampling method used here failed to fully capture sesquiterpenes (Helmig et al., 2004; Frazier et al., 2022). In addition, the lower molecular weight species (< $C_2$) such as methanol could not be characterized by a pre-concentrator coupled with the GC-MSD technique.*" (Line 239-244)

**References**

Helmig, D., Bocquet, F., Pollmann, J., Revermann, T., 2004. Analytical techniques for sesquiterpene emission rate studies in vegetation enclosure experiments. *Atmos. Environ.* 38, 557-572. https://doi.org/10.1016/j.atmosenv.2003.10.012.

Frazier, G., McGlynn, D.F., Barry, L.E., Lerdau, M., Pusede, S.E., Isaacman-VanWertz, G., 2022. Composition, concentration, and oxidant reactivity of sesquiterpenes in the southeastern U.S. *Environ. Sci.: Atmos.* 2, 1208-1220. https://doi.org/10.1039/d2ea00059h.

- l. 232: The authors lead the discussion of seasonal pattern with a figure in the supplement material (Fig. S4). If the figure is important, it should be part of the main text. Also, I wonder if Fig. S4 is absolutely necessary given that the authors should be able to support their conclusions with Fig. 3 from the main text (mentioned later in this section), as well as with Fig. 5.

**Reply:** Thank you for the suggestions. We agreed with your comments and deleted Fig. S4 as you suggested.

- Conclusions: I would like to again mention that it would be interesting and important for context to give the reader some information about the amount of litter naturally occurring at the wetland, potential changes in litter amount throughout the seasons, and how litter amounts are expected to change in a warming climate? Is more or less litter, expected to be found in the wetland? This information might also be included in the introduction and the discussion sections of the manuscript.

**Reply:** Thanks. As replied to the comments above, on the one hand we are currently unable to estimate the amount of litter naturally occurring in the wetland; On the other hand, estimating naturally occurring litter is beyond the scope of this manuscript. We greatly appreciate your valuable comments and suggestions. In the revised manuscript, we added some sentences on plant litter naturally occurring in wetland in the *Introduction*, which were also provided as follows:

"*Plant litter includes dead plants and dead plant material detached from living plants, the amount of which is significantly affected by climate (Cornwell et al., 2008). Future*

*global warming could alter the growth cycle of plants and accelerate plant litter breakdown, potentially leading to more litter BVOC emission.*" (Line 69-73)

**Reference**

Cornwell W.K., Cornelissen J.H., Amatangelo K., Dorrepaal E., Eviner V.T., Godoy O., Hobbie S.E., Hoorens B., Kurokawa H., Pérez-Harguindeguy N., Quested H.M., Santiago L.S., Wardle D.A., Wright I.J., Aerts R., Allison S.D., van Bodegom P., Brovkin V., Chatain A., Callaghan T.V., Díaz S., Garnier E., Gurvich D.E., Kazakou E., Klein J.A., Read J., Reich P.B., Soudzilovskaia N.A., Vaieretti M.V., Westoby M., 2008. Plant species traits are the predominant control on litter decomposition rates within biomes worldwide. *Ecol Lett.* 11, 1065-1071. https://doi.org/10.1111/j.1461-0248.2008.01219.x.

- Text S1: The authors seem to describe their laboratory blanks, but then mention that 'only when no targeted VOC were detected' the canisters were used for sampling. It is a little confusing. Why is only 10% of the evacuated canisters then analysed? Or have all the canisters been analysed earlier at some point for this project?

**Reply:** Thanks. Before sampling, the canisters must be cleaned to remove any possible residual contaminants. To check whether the canisters were really clean, we refilled the cleaned canisters with high pure $N_2$ and analyzed them by GC-MSD in the same way as the samples. We have agreed with you that all cleaned canisters should be checked, but this process is laborious, time-consuming and can be unnecessary. Because based on our 16 years of experience in VOC laboratory analysis, the canister cleaning method

we reported in the manuscript is reliable and no targeted VOC species are measured in the cleaned canisters. Thus, in the later experiments, we randomly selected 10% of the cleaned canisters for further check before sampling. If targeted VOCs are detected in any of the selected cleaned canisters, all cleaned canisters will be re-cleaned and inspected. This method for verification of canister cleanliness is also recommended by US EPA (https://www.epa.gov/sites/default/files/2019-12/documents/to-15a_vocs.pdf).

- Figure S2: I fail to understand the blue columns in this figure. I understand the numbers, but I don't understand their relation to the blue columns and if the blue columns have the same x-axis as the green bars. This should be made more understandable.

**Reply:** Sorry for that. Based on your comments, we have revised the Fig. S2 in the updated version.

[Figure]

**Fig. S2.** The carbon fluxes contributed by VOCs in the three treatments. Error bar was

the standard error. Red column represents the percentage of VOC-driven carbon in total calculated carbon (VOCs, $CO_2$ and $CH_4$).

Technical/language comments:

- l. 245-247: Should the sentence end with 'in all three treatments' or 'in the three treatments'?

Reply: Thanks. We revised as "in all three treatments".

"in all three treatments" (Line 260)

- l. 324: I would not use the turn of phrase 'ranked No. 1 contributor', which seems a little clunky compared to 'contributed most' or 'was the main contributor'.

Reply: Thank you. Revised as suggested.

"contributed most" (Line 338)

- ll. 417 and 427: The conclusions include twice the acronym TVOC, which has not been defined.

Reply: Thanks. We revised this in the manuscript.

"the net fluxes of total volatile organic compounds (TVOCs)" (Line 430)

- l. 430: While 'increasement' can be found in a dictionary, it is obsolete and can be replaced by 'increase'.

Reply: Revised as suggested.

"increase" (Line 443)

- l. 735: '11 campaigns': I would think that all the measurements in this study form a campaign, not each individual sampling time.

**Reply:** Thank you. We agreed with you and modified "11 campaigns" as "11 samplings".

"11 samplings" (Line 101-102, 731)

- Table S1: I noticed some small inconsistencies in the number of digits reported in this Table (e.g. sometimes ±0.1 and sometimes ±0.10).

**Reply:** Thanks. Based on your comments, we have revised the number of digits and kept it consistent in the Table S1.

---

## Author Comment (AC2)

**Community comments**:

Fang et al. provide an interesting study of the biogeochemistry of volatile organic compounds in a freshwater wetland. The study measured the air-wetland exchange of VOCs through year-long in situ field experiments and investigated the impact of plant litter decomposition on these measured VOCs. Overall, this manuscript is well organized and most of results are clearly explained, and it fits within the scope of Biogeosciences. However, some issues in the current manuscript still need to be addressed. Thus, I recommend it for publication after a minor revision.

**Reply:** Thank you for the helpful comments and providing us the opportunity to revise the manuscript. We have carefully addressed the comments in point-by-point form as shown below.

1. The abstract is well written but includes many abbreviations that are not needed in the abstract text and should be removed. Why do you include an abbreviation for a definition that you use one time only? These abbreviation explanations (NMHCs, OVOCs, VOSCs) should be moved to the manuscript text.

**Reply:** Revised as suggested. (Line 30, 32, 34)

2. The references are outdated, especially in the Introduction section. Please replace with more recent studies.

**Reply:** Revised as suggested.

3. In general, the experiment is designed comprehensively and the aim is clear. One question, however, is whether the authors measured the background VOCs of the chamber itself or whether these VOC concentrations were very low and could be neglected? I cannot find any relevant information in the manuscript.

**Reply:** Thanks for the comments. In fact, as you mentioned, this preliminary experiment was done. Based on your comments, we have added some detailed information about background VOCs of the chamber in the revision, which was also provided as follows:

"*A blank test was carried out by enclosing the chamber with the Teflon film. First, high-purity $N_2$ (99.999%) was used to purge the chamber and remove the ambient air in the chamber. After that, cleaned canisters were used to collect the first air sample from the chamber and then the second sample and third sample were collected after 30 and 45 minutes, respectively. During sampling, high-purity $N_2$ was gently added into the chamber to equalize the gas pressure. The target VOCs reported in this study were not detected.*" (Text S1)

**References**

Hornyák-Mester, E., Mentes, D., Farkas, L., Hatvani-Nagy, A.F., Varga, M., Viskolcz, B., Muránszky, G., Fiser, B., 2023. Volatile emissions of flexible polyurethane foams as a function of time. ***Polym. Degrad. Stab.*** 216, 110507. https://doi.org/10.1016/j.polymdegradstab.2023.110507.

Lattuati-Derieux, A., Thao-Heu, S., Lavédrine, B., 2011. Assessment of the degradation of polyurethane foams after artificial and natural ageing by using pyrolysis-gas

chromatography/mass spectrometry and headspace-solid phase microextraction-gas chromatography/mass spectrometry. *J. Chromatogr. A.* 1218, 4498-4508. https://doi.org/10.1016/j.chroma.2011.05.013.

4. I am a little interested in the AH emissions reported in the manuscript. As we know, AHs have traditionally been considered as anthropogenic VOCs. Are there any possible reasons or mechanisms for the emission of AHs from natural environments?

**Reply:** Yes. As you mentioned, studies on biogenic AH emission were quite limited. Because AHs are generally considered as anthropogenic pollutants. However, recent studies reported that biogenic emission could be a potential source of atmospheric AHs. For example, several studies found biogenic AH emissions in the decomposition of wheat straw, eutrophic lakes and oceanic environments (Rocco et al., 2021; Wohl et al., 2023; Wu et al., 2023; Fang et al., 2025).

The mechanisms of biogenic AH emission are very complex and are involved in many biogeochemical processes. Living plants can rapidly produce AHs and release them into the ambient air to protect against biotic and abiotic stresses (Misztal et al., 2015). Moreover, plant litter itself contains AHs when it falls to the ground and may release a certain number of AHs when broken down. Our previous study found that toluene emission during straw decomposition was about three times higher under flooded condition than under non-flooded condition, and was positively correlated with bacteria and fungus number, microbial biomass carbon, $CO_2$ flux under flooded condition, but their negative relationships were found under non-flooded condition. These results

reflect that toluene could be mainly produced by microorganisms during straw decomposition under anerobic or anoxic conditions (Wu et al., 2023).

In addition, different AH species might have different formation mechanisms. For example, Wohl et al. (2023) revealed that toluene and benzene measured in seawater had different biological sources. Rocco et al. (2021) also found that benzene strongly correlated with ethylbenzene and xylenes, but not with toluene in marine phytoplankton emission.

Overall, biogenic AH emissions do occur in natural environment and are generated through a series of biogeochemical processes that require more studies to explore and clarify.

**References:**

Misztal, P.K., Hewitt, C.N., Wildt, J., Blande, J.D., Eller, A.S.D., Fares, S., Gentner, D.R., Gilman, J.B., Graus, M., Greenberg, J., Guenther, A.B., Hansel, A., Harley, P., Huang, M., Jardine, K., Karl, T., Kaser, L., Keutsch, F.N., Kiendler-Scharr, A., Kleist, E., Lerner, B. M., Li, T., Mak, J., Nölscher, A.C., Schnitzhofer, R., Sinha, V., Thornton, B., Warneke, C., Wegener, F., Werner, C., Williams, J., Worton, D.R., Yassaa, N., Goldstein, A.H., 2015. Atmospheric benzenoid emissions from plants rival those from fossil fuels. *Sci. Rep.* 5, 12064. https://doi.org/10.1038/srep12064.

Rocco, M., Dunne, E., Peltola, M., Barr, N., Williams, J., Colomb, A., Safi, K., Saint-Macary, A., Marriner, A., Deppeler, S., Harnwell, J., Law, C., Sellegri, K., 2021. Oceanic phytoplankton are a potentially important source of benzenoids to the

remote marine atmosphere. ***Commun. Earth Environ.*** 2, 1–8. https://doi.org/10.1038/s43247-021-00253-0.

Wohl, C., Li, Q.Y., Cuevas, C.A., Fernandez, R.P., Yang, M.X., Saiz-Lopez, A., Simó, R., 2023. Marine biogenic emissions of benzene and toluene and their contribution to secondary organic aerosols over the polar oceans. ***Sci. Adv.*** 9, eadd9031. https://doi.org/10.1126/sciadv.add9031.

Wu, T., Zhao, X.Y., Liu, M.D., Zhao, J., Wang, X.M., 2023. Wheat straw return can lead to biogenic toluene emissions. ***J. Environ. Sci.*** 124, 281–290. https://doi.org/10.1016/j.jes.2021.08.050.

Fang, H., Wu, T., Ma, S.T., Miao, Y.Q., Wang, X.M., 2025. Biogenic emission as a potential source of atmospheric aromatic hydrocarbons: Insights from a cyanobacterial bloom-occurring eutrophic lake. ***J. Environ. Sci.*** 151, 497−504. https://doi.org/10.1016/j.jes.2024.04.011.

5. Line 179: volatile organic sulfide compounds (VOSCs)

**Reply:** Revised as suggested. (Line 190)

6. Line 227-228. Monoterpene can be detected by GC/MSD method (Yuan et al. 2023. Emissions of isoprene and monoterpenes from urban tree species in China and relationships with their driving factors). The canister sampling used in the experiment may not be appropriate to capture these higher-molecular-weight species.

**Reply:** Thank you for carefully reviewing our manuscript. As you stated, canister sampling combined with GC/MSD method can be used for analyzing monoterpenes. The higher molecular weight species mentioned in the manuscript referred to the sesquiterpenes. In the updated version, we have revised "monoterpenes" as "*sesquiterpenes*". (Line 242)

7. Line 398: Change abbreviations "VOSCs" as "Volatile organic sulfide compounds".

**Reply:** Revised as suggested. (Line 411)

---

## Author Comment (AC3)

**Reviewer #2**

Title: Litter decomposition enhances volatile organic compound emission from a freshwater wetland: insights from year-round in situ field experiments

Author(s): Hua Fang et al.

MS No.: egusphere-2023-2998

The authors report VOC fluxes from the decaying leaf litter of a freshwater wetland plant, Phragmites australis. The reported fluxes include an impressive list of 62 VOCs: 5 aromatics, 15 OVOCs, 30 NMHCs, 9 HVOCs, 3 VOSCs. Many of these compounds are difficult to quantify well, thus requiring proof of careful methodology. The proof, unfortunately, is not provided, leading to doubts about data quality. The problem is compounded by the use of incomplete references. The methods section cites two papers (Wang et al, 2023 and Liu et al. 2021) to broadly explain 'analysis steps'. The cited studies employ slightly different methods and themselves cite a range of earlier papers for broad methodological statements. It becomes a nearly impossible task to follow the actual methods of VOC quantification. Specific papers should be cited for

specific methods, and only if they truly align. For example, Wang et al., 2023, uses a different instrument altogether (GC-MSD-FID from Agilent vs. a GC-MSD from Shimadzu), so it should be made evident why this paper is cited. Below, I compile a checklist for methodology that should be included for field measurements of VOC fluxes. While several items are addressed in this study, other critical aspects are missing.

**Reply:** Thanks for the comments and suggestions. We are very sorry that Reviewer #2 was concerned about the methodology used in this study, which in fact has been evaluated and also

used in previous studies published in ***Biogeosciences.*** Below we tried to address your concerns and provided more detailed explanation.

Furthermore, litter BVOCs are diverse and complex, and we only reported 62 VOCs in this work. Well over 62 different VOCs have been reported in the previous studies. For example, Svendsen et al. (2018) reported 84 different VOCs emitted from the *Salix* litter and the similar semi-quantification method was used in their study.

As for the literatures cited, the main analysis steps of them were the same as ours. First, VOCs are concentrated by a pre-concentrator, and then are analyzed by GC-MSD. Although the instrument type was different between this work and Wang et al. (2023), the running procedures of the pre-concentrator and GC-MSD, such as temperature setting in each step, were the same. We did refer the studies to conduct the lab analysis and thus cited them. In addition, the lab analysis described in the manuscript was also clear enough for readers to understand.

Anyway, based on your comments, we have deleted the cited references in original manuscript and modified the description on laboratory analysis.

**References**

Svendsen, S.H., Priemé, A., Voriskova, J., Kramshøj, M., Schostag, M., Jacobsen, C.S., Rinnan, R., 2018. Emissions of biogenic volatile organic compounds from arctic shrub litter are coupled with changes in the bacterial community composition. Soil Biology and Biochemistry. 120, 80-90. https://doi.org/10.1016/j.soilbio.2018.02.001.

**Methodology**

**Sample storage:**

**Canister stability**

Stability of compounds was not tested for the compounds of interest, even though they have different degrees of stability in canisters. Compounds with poor stability or even production within canisters from other compounds (e.g., VOSCs) need to be checked.

**Reply:** Thanks for the comments. Canister sampling for VOCs was widely used in field observations and recommended by US EPA. The method used here for collecting VOCs referred to the previous chamber-based flux studies (Hellén et al., 2006; Whelan and Rhew, 2016) and the method of PAMS and TO-15, which reported that under conditions of normal usage for sampling ambient air, most VOCs can be recovered from canisters near their original concentrations after storage times of up to thirty days (https://www.epa.gov/sites/default/files/2019-11/documents/to-15r.pdf, *Section 1.3*). Haapanala et al. (2006) reported that 24 different NMHCs and 7 halocarbons can preserve in the canisters for at least one week. In fact, US EPA has sponsored the investigation of the SilcoSteel™ process of coating the canister interior with a film of fused silica to reduce surface activity. Of course, it should be admitted that humidity could influence the storage time of VOCs in canister. For example, some VOCs are not stable for more than a week under 50% of relative humidity. In addition, Guo et al. (2010) also reported that DMS collected in canisters was stable for a week. Yu et al. (2024) reported that VSCs (DMS, COS and $CS_2$) were stable for sixteen-days storage in canisters at room temperature. However, our samples collected by silonite-treated stainless steel canisters were immediately transported to the laboratory after

sampling and were analyzed within two days.

**References**

Yu, J., Yu, L., He, Z., Yang, G.P., Lai, J.G., Liu, Q., 2024. Spatial and seasonal variability in volatile organic sulfur compounds in seawater and the overlying atmosphere of the Bohai and Yellow seas. *Biogeosciences.* 21, 161-176. https://doi.org/10.5194/bg-21-161-2024.

Guo, H., Simpson, I.J., Ding, A.J., Wang, T., Saunders, S.M., Wang, T.J., Cheng, H.R., Barletta, B., Meinardi, S., Blake, D.R., Rowland, F.S., 2010. Carbonyl sulfide, dimethyl sulfide and carbon disulfide in the Pearl River Delta of southern China: impact of anthropogenic and biogenic sources. *Atmos. Environ.* 44, 3805-3813. https://doi.org/10.1016/j.atmosenv.2010.06.040.

Haapanala, S., Rinne, J., Pystynen, K. H., Hellén, H., Hakola, H., Riutta, T., 2006. Measurements of hydrocarbon emissions from a boreal fen using the REA technique. *Biogeosciences.* 3, 103-112. https://doi.org/10.5194/bg-3-103-2006.

Hellén, H., Hakola, H., Pystynen, K.H., Rinne, J., Haapanala, S., 2006. $C_2$-$C_{10}$ hydrocarbon emissions from a boreal wetland and forest floor. *Biogeosciences.* 3, 167–174. https://doi.org/10.5194/bg-3-167-2006.

Whelan, M.E., Rhew, R.C., 2016. Reduced sulfur trace gas exchange between a seasonally dry grassland and the atmosphere. *Biogeochemistry*. 128, 267–280. https://doi.org/10.1007/s10533-016-0207-7.

**Field sampling:**

**Flux chamber blank (sampling equipment only)**

Chamber blank/control experiments using empty chambers were not conducted, which is concerning because of the use of foam board to make the chamber float. Separate issue: chambers are described as 1.1m x 1.1m x 1.1m (Line124), but it appears that another smaller chamber (35 x 35 x 15cm) is then placed inside the larger box (Line 140), is that correct?

**Reply:** Thanks for the comments.

1) In fact, we also considered this issue when conducting the experiments. Therefore, the foam board was completely wrapped with Teflon film and installed around the chamber. In addition, previous studies have already reported that most of the VOCs released from urethane foam were siloxane-type compounds and the VOCs reported in our study were not detected. (Lattuati-Derieux et al., 2011; Hornyák-Mester et al., 2023). Moreover, the chamber blank experiment was actually conducted before formal sampling. Based on your comments, we have added the relevant information in *Supplement* as below:

"*A blank test was carried out by enclosing the chamber with the Teflon film. First, high-purity $N_2$ (99.999%) was used to purge the chamber and remove the ambient air in the chamber. After that, cleaned canisters were used to collect the first air sample from the chamber and then the second sample and third sample were collected after 30 and 45 minutes, respectively. During sampling, high-purity $N_2$ was gently added into the chamber to equalize the gas pressure. The target VOCs reported in this study were not detected.*" (Text S1)

2) Yes. Nine 1.1m × 1.1m × 1.1m stainless steel cuboidal boxes without the cover were installed in the freshwater wetland. Three treatments (A: no litter; B: 1.4 kg litters; C: 2.8 kg litters) were set and each of the treatment consisted of three parallel groups, as shown in the Fig.1 in the

manuscript. The static chamber (35cm × 35 cm × 15cm) used for flux measurement were put inside the cuboidal boxes.

**References**

Hornyák-Mester, E., Mentes, D., Farkas, L., Hatvani-Nagy, A.F., Varga, M., Viskolcz, B., Muránszky, G., Fiser, B., 2023. Volatile emissions of flexible polyurethane foams as a function of time. *Polym. Degrad. Stab.* 216, 110507. https://doi.org/10.1016/j.polymdegradstab.2023.110507.

Lattuati-Derieux, A., Thao-Heu, S., Lavédrine, B., 2011. Assessment of the degradation of polyurethane foams after artificial and natural ageing by using pyrolysis-gas chromatography/mass spectrometry and headspace-solid phase microextraction-gas chromatography/mass spectrometry. *J. Chromatogr. A.* 1218, 4498-4508. https://doi.org/10.1016/j.chroma.2011.05.013.

**Leakage / sample replacement**

A correction is needed to account for the removal of 3.2 L samples taken from an 18 L chamber. This is not a large correction, given that the first sample is taken at the beginning when outside air = inside air, but it should be considered (if using 3 points instead of 2). Leakage/sample replacement may be responsible for the non-linearity of concentration change that was observed.

**Reply:** Thanks. In this work, two-point flux calculation was used, which was also suggested and used in the previous studies (Hellén et al., 2006; Maier et al., 2022). The collection of a 3.2 L sample accounted for < 20% of the chamber and we therefore did not correct for the effects of sample replacement. It is acknowledged that this manual sampling may introduce uncertainty

in the actual flux estimate. However, this sampling behavior remained consistent across the three treatments and 11 samplings in this work, allowing the results to be compared.

**References**

Hellén, H., Hakola, H., Pystynen, K.H., Rinne, J., Haapanala, S., 2006. $C_2$-$C_{10}$ hydrocarbon emissions from a boreal wetland and forest floor. *Biogeosciences.* 3, 167–174. https://doi.org/10.5194/bg-3-167-2006.

Maier, M., Weber, T.K.D., Fiedler, J., Fuß, R., Glatzel, S., Huth, V., Jordan, S., Jurasinski, G., Kutzbach, L., Schäfer, K., Weymann, D., Hagemann, U., 2022. Introduction of a guideline for measurements of greenhouse gas fluxes from soils using non-steady-state chambers. *J. Plant Nutr. Soil Sci.* 185, 447–461. https://doi.org/10.1002/jpln.202200199.

**Instrument:**

Hardware and chromatography details. Yes. Entech pre-concentrator with Shimadzu GC-MSD. Temperature programming. Note that running in SCAN mode is less precise than SIM (Lines 156-170).

**Reply:** Thanks for the comments. For GC/MSD, SIM mode does offer better sensitivity and selectivity than SCAN mode. SIM mode is very convenient when only a select number of analytes are important for the study and all other compounds present in the sample can be ignored. However, if we have no preconceived notion about what might be present in the environmental samples, as shown in this work and previous studies on BVOCs (Leff and Noah, 2008; Ryde et al., 2022), a SCAN mode is therefore appropriate (Geer Wallace et al., 2017).

**References**

Leff, J. W., and Fierer, N., 2008. Volatile organic compound (VOC) emissions from soil and litter samples. *Soil Biol. Biochem.* 40, 1629–1636, https://doi.org/10.1016/j.soilbio.2008.01.018.

Ryde, I., Davie-Martin, C.L., Li, T., Naursgaard, M.P., Rinnan, R., 2022. Volatile organic compound emissions from subarctic mosses and lichens. *Atmos. Environ.* 290, 119357. https://doi.org/10.1016/j.atmosenv.2022.119357.

Geer Wallace, M.., Pleil, Joachim D., Mentese, S., Oliver, K.D., Whitaker, D.A., Fent, K.W., 2017. Calibration and performance of synchronous SIM/scan mode for simultaneous targeted and discovery (non-targeted) analysis of exhaled breath samples from firefighters. *J. Chromatogr. A.* 1516, 114-124. https://doi.org/10.1016/j.chroma.2017.07.082.

Instrument precision. Not reported. Precisions are needed for each compound.

**Reply:** We added this in the updated version. (Text S1)

"*Text S1 Quality assurance and quality control*

*Before sampling, the canisters were repeatedly filled and evacuated pure nitrogen at least three times to remove the potential contaminants and then the evacuated canisters were placed in laboratory for 24 hours. After that, 10% evacuated canisters were selected randomly to be refilled with pure nitrogen and were analyzed in the same way as the samples. Only when no targeted VOCs were detected, the canisters were considered as clean ones and can be used for formal sampling.*

*A blank test was carried out by enclosing the chamber with the Teflon film. First, high-purity $N_2$ (99.999%) was used to purge the chamber and remove the ambient air in the chamber. After that, cleaned canisters were used to collect the first air sample from the chamber and then the*

*second sample and third sample were collected after 30 and 45 minutes, respectively. During sampling, high-purity N$_2$ was gently added into the chamber to equalize the gas pressure. The target VOCs reported in this study were not detected.*

*Field blank canisters refilled with pure nitrogen were brought to sampling site and returned to laboratory (lab) for analysis in the same way as samples. The targeted VOCs were not detected or presented the level below the method detection limits (MDLs). During the period of lab analysis, lab blank (the canister filled with pure nitrogen) was analyzed firstly to check if any contaminants remained in the system of GC/MSD coupled with pre-concentrator. All VOCs detected in this work were identified based on their retention times (RTs) in GC and m/z obtained from MSD and were quantified by calibration curves. The VOCs without standards were identified by their match to the NIST library and were semi-quantified based on the species whose RTs were close to them or based on the species which have a similar chemical structure to them. The gas standards were prepared by dynamically diluting the Photochemical Assessment Monitoring Stations (PAMS) standard mixture and TO-15 standard mixture (Linde Spectra Environment Gases, USA) to 0.5, 1, 5, 15 and 30 ppb, respectively. The pure liquid standards of dimethyl sulfide (DMS) and dimethyl disulfide (DMDS) were purchased from Sigma-Aldrich and were dynamically diluted to 0.1, 0.5, 1, 2 and 4 ppb, respectively. The calibration curves were obtained by running the five diluted standards, plus the humidified zero air, in the same manner as the collected samples. Each day before the sample analysis, the system was checked by high pure nitrogen (99.999%) and calibrated with a one-point calibration (1ppb). If the relative percent difference from the initial calibration curve were > 15%, the recalibration is made. For calibrated compounds, the analytical precision determined*

*as a standard deviation of the calibration standards (n=7) was < 12%. The MDLs of these calibrated VOCs ranged from 0.001 ppb to 0.040 ppb. Further, the minimum detectable flux (MDF) of the chamber for VOCs was estimated based on their MDLs (Pihlatie et al., 2013). By assuming a minimum increase of VOC concentration within the chamber headspace equal to its MDL during the 10 mins of chamber enclosure. The resulting MDFs for chamber ranged from 0.002 $\mu g\ m^{-2}\ h^{-1}$ to 0.074 $\mu g\ m^{-2}\ h^{-1}$.*" (Text S1)

Duplicate runs. Not reported.

**Reply:** We added this in the updated version. (Text S1)

Detection limit. Not reported.

**Reply:** We added this in the updated version. (Text S1)

Use of Gas standards. Diluted PAMS and TO-15 standard mixtures are cited. DMS and DMDS standards were used but not described. For VOCs without standards, they were "semi-quantified based on the species whose RTs were close to them or based on the species which have a similar chemical structure to them." This is not an appropriate method, as there is no reason to believe the instrument sensitivity or ion fragmentation will be the same. These compounds must be removed from any quantitative analysis.

**Reply:** Thanks for the comments. We agreed with you that an appropriate quantification of VOCs ideally requires the use of specific standards for the VOCs presented in samples. However, this manuscript aims to explore what kind of VOCs involved in the exchange process

between wetland and atmosphere, the standards for each compound measured here are thus not known a priori. Other method that allows semi-quantification of a compound using the nearest or similar components within the chromatogram can be used. Previous studies on BVOCs also met similar problems and the semi-quantification method was commonly used by them (Frazier et al., 2022; Ruiz-Hernández et al., 2018; Ryde et al., 2022), and some of these studies were also published in *Biogeosciences* (Brachmann et al., 2023; Jaakkola et al., 2023; Kleist et al., 2012; van Meeningen et al., 2016, 2017).

**References**

Ruiz-Hernández, V., Roca, M.J., Egea-Cortines, M., Weiss, J., 2018. A comparison of semi-quantitative methods suitable for establishing volatile profiles. *Plants Methods.* 14, 67. https://doi.org/10.1186/s13007-018-0335-2.

Ryde, I., Davie-Martin, C.L., Li, T., Naursgaard, M.P., Rinnan, R., 2022. Volatile organic compound emissions from subarctic mosses and lichens. *Atmos. Environ.* 290, 119357. https://doi.org/10.1016/j.atmosenv.2022.119357.

Frazier, G., McGlynn, D.F., Barry, L.E., Lerdau, M., Pusede, S.E., Isaacman-VanWertz, G., 2022. Composition, concentration, and oxidant reactivity of sesquiterpenes in the southeastern U.S. *Environ. Sci.: Atmos.* 2, 1208-1220. https://doi.org/10.1039/d2ea00059h.

Brachmann, C.G., Vowles, T., Rinnan, R., Björkman, M. P., Ekberg, A., Björk, R.G., 2023. Herbivore–shrub interactions influence ecosystem respiration and biogenic volatile organic compound composition in the subarctic. *Biogeosciences*. 20, 4069-4086. https://doi.org/10.5194/bg-20-4069-2023.

Jaakkola, E., Gärtner, A., Jönsson, A.M., Ljung, K., Olsson, P.O., Holst, T., 2023. Spruce bark beetles (Ips typographus) cause up to 700 times higher bark BVOC emission rates compared to healthy Norway spruce (Picea abies). *Biogeosciences.* 20, 803-826. https://doi.org/10.5194/bg-20-803-2023.

Kleist, E., Mentel, T. F., Andres, S., Bohne, A., Folkers, A., Kiendler-Scharr, A., Rudich, Y., Springer, M., Tillmann, R., Wildt, J., 2012. Irreversible impacts of heat on the emissions of monoterpenes, sesquiterpenes, phenolic BVOC and green leaf volatiles from several tree species. *Biogeosciences.* 9, 5111-5123. https://doi.org/10.5194/bg-9-5111-2012.

van Meeningen, Y., Schurgers, G., Rinnan, R., Holst, T., 2017. Isoprenoid emission response to changing light conditions of English oak, European beech and Norway spruce. *Biogeosciences.* 14, 4045-4060. https://doi.org/10.5194/bg-14-4045-2017.

van Meeningen, Y., Schurgers, G., Rinnan, R., Holst, T., 2016. BVOC emissions from English oak (Quercus robur) and European beech (Fagus sylvatica) along a latitudinal gradient. *Biogeosciences.* 13, 6067-6080.   https://doi.org/10.5194/bg-13-6067-2016.

Gas standard traceability. Missing. Need to add source of standard and standard traceability (if available).

**Reply:** We added this in the updated version.

"*Photochemical Assessment Monitoring Stations (PAMS) standard mixture and TO-15 standard mixture (Linde Spectra Environment Gases, USA)*" (Text S1)

"*The pure liquid standards of dimethyl sulfide (DMS) and dimethyl disulfide (DMDS) were purchased from Sigma-Aldrich and were dynamically diluted to 0.1, 0.5, 1, 2 and 4 ppb*" (Text

S1)

Cal Curve over range of concentrations. May not be valid, as VOC concentrations / ranges were not reported. Calibration curves might not be appropriate for the range of concentrations observed. If concentrations < 0.5 ppb (or <100 ppt for VOSCs), then the calibration curve is a one point calibration.

**Reply:** Thanks. There might be some misunderstanding in description of establishing calibration curves. In fact, our calibration curves included the origin which used the humidified zero air. The following figure shows the calibration curves of typical VOCs measured in this work. Thus, the VOC concentrations above the method detected limits can be quantified by the calibration curves ($R^2 > 0.99$). We modified the description on establishing the calibration curves, which was provided as below:

"*The calibration curves were obtained by running the five diluted standards, plus the humidified zero air, in the same manner as the collected samples.*" (Text S1)

[Figure]

**Flux calculation**

Curve fitting procedures. "VOC concentrations in chamber headspace had already leveled off at 30 min. Thus, here the first two points, which could capture the initial fluxes, were used to calculate VOC fluxes (Zhang et al., 2021)." What is meant by 'leveled off', and do all chambers show this trend? Zhang et al. only used 2 point flux calculations for one compound and only in the <10% of cases when r2<0.75. Here, the 2-point method appears to be used for all VOCs. Is the system leaky or is there a dilution effect from withdrawing large samples (see above)?

**Reply:** Thanks for the comments. In fact, the VOC concentrations leveling off in the chamber headspace was not a phenomenon unique to this work, but a common problem with static chamber approach. The VOC concentrations in the chamber headspace increase asymptotically during the sampling period and it leads to very high VOC accumulation in the chamber headspace, which in turn inhibits the continued increase in headspace VOC concentration as wetland to atmosphere gradient is altered. Gao and Yates (1998) have already reported this phenomenon (see Figure 2 in their article).

Due to VOC accumulation within the chamber headspace, the flux measured by a static chamber during its placement underestimates the actual flux. The longer the chamber placement time, the more severe the underestimate is (Gao and Yates, 1998). Therefore, the VOC concentrations in the chamber headspace might exhibit a nonlinear increase with time, particularly over a long sampling period. This is the reason why many studies used the initial stage of chamber placement to calculate gas flux, in which the target gas concentrations are more likely to increase linearly with time (Heinemeyer and McNamara, 2011). This to some degree prevents flux underestimation due to a build up of headspace VOC concentrations.

The flattening of headspace VOC concentration was not observed in all chambers in this work. We used the first two points to calculate flux to keep VOC flux calculations consistent in three treatments. In this way, we can compare the VOC fluxes in different treatments and in different seasons. Similarly, this method to calculate BVOC fluxes was also used by Hellén et al. (2006). We acknowledged that this could induce uncertainty in calculating VOC fluxes. However, if the third point (30 min) was included, VOC fluxes could be considerably underestimated as suggested by Gao and Yates (1998). In addition, although Zhang et al. (2021) employed two-point flux calculations only in the < 10% of cases, this still reflects that the method could capture the initial gas fluxes and the calculated results were acceptable. If not, they could have excluded the flux data directly instead of continuing to report the results in their study.

Anyway, we greatly appreciate your comments and suggestions. In the revised manuscript, we added some sentences as follows:

"*In fact, VOC concentration in chamber headspace has already leveled off at 30 minutes in some cases, which could be because the accumulation of VOCs in the chamber significantly decreases the natural VOC concentration gradient between wetland and atmosphere (Heinemeyer and McNamara, 2011). It was a common problem for static chamber approach. Previous studies suggested shortening the sampling time or using the initial stage of chamber placement to calculate the gas fluxes (Gao and Yates, 1998; Hellén et al., 2006; Heinemeyer and McNamara, 2011; Silva et al., 2015; Zhang et al., 2021). In addition, to compare the VOC fluxes measured in different treatments and in different seasons, the first two points, which could capture the initial fluxes, were thus used to calculate VOC fluxes.*" (Line 156-165)

"*Furthermore, online measurements can obtain more VOC concentration data in the initial*

*stage of chamber placement, reducing the uncertainty in flux calculation, as shown in the present study*." (Line 456-458)

**References**

Gao, F., Yates, S.R., 1998. Laboratory study of closed and dynamic flux chambers: experimental results and implications for field application. *J. Geophys. Res-Atmos.* 103, 26115–26125. https://doi.org/10.1029/98JD01346.

Heinemeyer, A., McNamara, N.P., 2011. Comparing the closed static versus the closed dynamic chamber flux methodology: implications for soil respiration studies. *Plant Soil*. 346, 145–151. https://doi.org/10.1007/s11104-011-0804-0.

Hellén, H., Hakola, H., Pystynen, K.H., Rinne, J., Haapanala, S., 2006. $C_2$-$C_{10}$ hydrocarbon emissions from a boreal wetland and forest floor. *Biogeosciences.* 3, 167–174. https://doi.org/10.5194/bg-3-167-2006.

Zhang, W., Jiao, Y., Zhu, R., Rhew, R.C., Sun, B., Dai, H., 2021. Chloroform ($CHCl_3$) emissions from coastal Antarctic tundra. *Geophys. Res. Lett.,* 48, e2021GL093811. https://doi.org/10.1029/2021GL093811.

Flux detection limits. Given the precision of each compound, what are their flux detection limits? When is a flux determined to be not significantly different from zero? How are these observations treated?

**Reply:** Thanks for the comments. Based on your comments, we have added the flux detection limits in the updated version (Text S1). In addition, the VOC fluxes were considered here when they were greater than the minimum detectable flux (MDF), which was used for the zero flux

test.

"*Further, the minimum detectable flux (MDF) of the chamber for VOCs was estimated based on their MDL (Pihlatie et al., 2013). Assuming a minimum increase of VOC concentration within the chamber headspace equal to its MDL during the 10 minutes of chamber enclosure. The resulting MDFs for chamber ranged from 0.002 to 0.074 μg m$^{-2}$ h$^{-1}$.*" (Text S1)

**References**

Pihlatie, M.K., Christiansen, J.R., Aaltonen, H., Korhonen, J.F.J., Nordbo, A., Rasilo, T., Benanti, G., Giebels, M., Helmy, M., Sheehy, J., Jones, S., Juszczak, R., Klefoth, R., Lobo-do-Vale, R., Rosa, A.P., Schreiber, P., Serça, D., Vicca, S., Wolf, B., Pumpanen, J., 2013. Comparison of static chambers to measure $CH_4$ emissions from soils. *Agric. For. Meteorol.* 171-172, 124-136. https://doi.org/10.1016/j.agrformet.2012.11.008.

In summary, numerous revisions are needed to address the methodological omissions and to improve the quality of flux data. Measuring 62 VOC fluxes is impressive, but how many of these fluxes are real? I suspect several of the measurements have larger uncertainties than presented and/or are below detection levels. Also, aggregating fluxes by VOC type can summarize a large set of results, but this process obscures what is happening with specific compounds. By aggregating by mass, heavier compounds can dominate the overall picture. Please provide more information about the measurements that are defensible, add caveats where needed, and exclude results that are unreliable. Only then can the reader be fully able to assess the scientific contributions of the work.

**Reply:** Thanks for the comments and suggestions. We acknowledged that, as in many static

chamber studies, there are some uncertainties in this work that could lead to the fluctuation in the VOC fluxes calculated here. However, the variation trend of wetland-atmosphere exchange of VOCs, such as the uptake or release of VOCs and the relationship between VOC flux and temperature, as well as the impacts of litter decomposition on VOCs in freshwater wetlands, would not change. Based on your comments and suggestions, we have added descriptions and explanations of these uncertainties to provide our readers with a clearer understanding and to enable subsequent researchers to conduct their studies accordingly.